# A Microwave Pressure Sensor Loaded with Complementary Split Ring Resonator for High-Temperature Applications

**DOI:** 10.3390/mi14030635

**Published:** 2023-03-10

**Authors:** Libo Yang, Hairong Kou, Xiaoli Wang, Xiaoyong Zhang, Zhenzhen Shang, Junbing Shi, Guanghua Zhang, Zhiguo Gui

**Affiliations:** 1Shanxi Provincial Key Laboratory of Intelligent Sensing and Multi-Dimensional Information Processing, Taiyuan University, Taiyuan 030032, China; 2Jinxi Industries Group Co., Ltd, Taiyuan 030024, China; 3Shanxi Provincial Key Laboratory for Biomedical Imaging and Big Data, North University of China, Taiyuan 030051, China

**Keywords:** pressure sensor, CSRR, harsh environment, HTCC

## Abstract

A passive substrate integrated waveguide (SIW) sensor based on the complementary split ring resonator (CSRR) is presented for pressure detection in high-temperature environments. The sensor pressure sensing mechanism is described through circuit analysis and the electromagnetic coupling principle. The pressure sensor is modeled in high frequency structure simulator (HFSS), designed through parameter optimization. According to the optimized parameters, the sensor was customized and fabricated on a high temperature co-fired ceramic (HTCC) substrate using the three-dimensional co-fired technology and screen-printing technology. The pressure sensor was tested in the high-temperature pressure furnace and can work stably in the ambient environment of 25−500 °C and 10−300 kPa. The pressure sensitivity is 139.77 kHz/kPa at 25 °C, and with increasing temperature, the sensitivity increases to 191.97 kHz/kPa at 500 °C. The temperature compensation algorithm is proposed to achieve accurate acquisition of pressure signals in a high-temperature environment.

## 1. Introduction

Pressure measurement in high-temperature environments [1,2,3] is of great importance in the fields of aerospace [4,5], mining and metallurgy [6,7], etc. During the operation of aerospace vehicles, the surface, engine, and gas turbine are usually accompanied by high temperature, high pressure, and other harsh environments. Therefore, it is necessary to accurately obtain the real-time pressure parameter to provide the premise and guarantee for the surface material selection, structural design, and protective measures of aerospace vehicles. For example, when the hypersonic vehicle runs at high speed, the friction between the surface structure and the atmosphere generates a lot of heat. Accurate acquisition of the mechanical and thermal parameters of the surface is especially essential for the design and protection of the surface structure.

At present, there are three main methods used for pressure detection in a high-temperature environments such as piezoresistive [8,9,10], fiber optic sensing [11,12,13], and microwave technology [14,15,16]. Conventional piezoresistive pressure sensors are difficult to apply in high-temperature environments (>500 °C) due to the transmission principle, material characteristics (polymer [17]), and electronic interconnection, which limit their development in high-temperature fields. The Fiberoptic Fabry–Perot pressure sensors also have broad applications in high-temperature environments owing to their advantages, such as compact sizes, high sensitivity, and high-temperature resistance. However, the Fabry–Perot pressure cavity is assembled with the optical fiber by using the fusion glue, which will be softened in a high-temperature environment, resulting in the optical fiber falling off. Microwave sensors have attracted growing attention for high-temperature applications due to their advantages of passivity, high precision, high sensitivity, and stable signal transmission.

Microwave technology was demonstrated for applying pressure detection [18,19,20,21]. Shamabadi et al. [18] developed a microwave pressure sensor, based on variations in the transmission zero of a microstrip circuit, which shows an excellent linear variation of transmission zero with a pressure sensitivity of 0.57 MHz/kPa when a pressure changes from 0 to 466 kPa. Mrozowski et al. [20] reported a microwave pressure sensor, based on microstrip line-fed ring resonators, which can measure an applied pressure up to 40.5 kPa with an average sensitivity of 9.62 kHz/Pa. Chen et al. [21] designed a microwave wire-interrogated pressure sensor, composed of a coaxial-like cavity resonator and microwave waveguide, which can work with a temperature range of 20−400 °C with a pressure sensitivity of 4.3 MHz/kPa. It is a pity that this sensor uses the traditional evanescent-mode resonator, which increases the volume of the sensor. Split-ring resonator (SRR) and complementary split-ring resonator (CSRR) have been widely integrated in microwave sensors for the measurement of permittivity [22,23], cracks [24,25], thickness [26], and glucose [27], etc. The SRR and CSRR have the advantages of compact structure, low loss, and multiband, which become an alternative to an evanescent-mode resonators. In [28], an integrated waveguide resonator in half mode of the evanescent mode substrate is designed for a wireless passive-sensing application. However, this structure and the wireless transmission method do not apply to the detection of pressure signals under a high-temperature environment. Wireless pressure detection method [29] is considered as the most feasible method to realize in situ pressure detection in high temperature environments because it is not subject to the failure of heterogeneous connection and the deterioration of conductivity caused by high temperature environments. However, there are new problems in wireless pressure signal transmission. For example, the ambient noise interferes and submerges the working signal, especially in metal containers, and the coupling energy dissipation in high temperature environments causes the close wireless interrogation distance.

In this paper, we propose a passive substrate integrated waveguide (SIW) pressure sensor based on CSRR, which realizes sensing applications in high-temperature environment at the microwave frequency band by using the electromagnetic coupling principle. A small sealing cavity, prepared by the HTCC three-dimensional integrated manufacturing process, is placed under the CSRR. When the sealing cavity is deformed under pressure, the electromagnetic field close to the CSRR resonator is disturbed, resulting in the shift of the resonant frequency. By analyzing the return loss (S11) of the resonant frequency of the sensor, the relationship between the environment and the sensor can be obtained. The sensor was simulated to operate at 1.6 GHz with high performance by HFSS. Finally, the sensor is tested on the high-temperature pressure composite test platform, which verifies that the sensor can be used to detect the pressure parameter in a high-temperature environment. In this scheme, we use CSRR resonator as a pressure-sensitive element and microwave transmission line as pressure signal transmission carrier. This scheme uses dielectric and microwave transmission lines for electromagnetic wave transmission, eliminates the current dependence of signals in wireless transmission, and avoids signal dissipation, depletion, and avalanche at high-temperature environments.

## 2. Sensor Design and Fabrication

### 2.1. Sensing Theory of Sensor

A schematic drawing of the CSRR pressure is presented in Figure 1. The sensor is composed of a SIW microwave cavity, a metal radiation patch with a CSRR structure, and an HTCC substrate with a sealed cavity. The HTCC substrate with a sealed cavity is manufactured using the HTCC three-dimensional integrated manufacturing process with 99% HTCC green tapes. The SIW structure, composed of parallel metal surfaces and metalized holes embedded in the substrate, has the advantages of a high Q factor, low environmental interference, and low insertion loss. A CSRR structure is embedded in the metal patch directly above the sealed cavity, using the metal line for signal transmission and serving as the first highly reflective reflector. The open end of metal line connected to the network analyzer acts as an open circuit and serves as the second highly reflective reflector. Consequently, a microwave SIW resonator based on CSRR is formed. The incident probing signal will be partially reflected by the CSRR structure; the transmitted portion continues traveling to the open end and is reflected at the open end of the metal line. Multiple reflections occur between the two reflectors, resulting in an obvious peak in the S11 curve, that is, the resonant frequency of the sensor.

The sealed cavity is the key sensitive element of the pressure sensor. First, the mechanical characteristics of the HTCC substrate with the sealed cavity are analyzed, and the COMSOL simulation software is used to simulate the pressure applied to the substrate. Figure 2a shows the substrate with clamped four edges and bottom plate under uniformly distributed pressure P (300 kPa). The mesh size option is normal size. The stress distribution of the substrate is shown in Figure 2b. It can be seen from Figure 2b that when the external pressure changes, the pressure-sensitive membrane of the sealed cavity will deform, and the maximum deformation of the pressure-sensitive membrane is d, which can be expressed as follows:(1)$$d=\frac{3P^{4}\left(1-\nu^{2}\right)}{16{\mathrm{Ea}}^{3}}$$
where ν and E represent Poisson ratio and Young’s modulus of the substrate material, P is the loaded pressure, and a is the thickness of the sensitive film.

According to the pressure value of the measured environment, we can improve the sensitivity of the sensor by reducing the thickness of the cavity sensitive film. At the same pressure, the thin sensitive film will increase the sink distance of the cavity, resulting in a larger shift of the resonant frequency of the sensor, thus improving the pressure sensitivity of the sensor.

Based on the circuit analysis method, the CSRR pressure sensor can be regarded as an equivalent circuit of the LC, as shown in Figure 2c. Here, C_1_ is the coupling capacitance between the two resonant rings in the CSRR structure, which can be ignored. The capacitance C is the difference between the upper and lower metal surfaces, and the inductance L is mainly determined by the resistance of the surface metal. According to the capacitance formula of parallel plates $C=\frac{\varepsilon_{0}\varepsilon_{r}A}{d}\;$(where $\varepsilon_{0}$ is the vacuum dielectric constant, $\varepsilon_{r}$ is the base dielectric constant, and A is the area of the upper and lower metal plates), when the ambient pressure changes, the sealed cavity will deform, the distance between the upper and lower plates will decrease, and the equivalent capacitance C in the equivalent circuit will increase. The resonant frequency of the CSRR pressure sensor can be expressed as Equation (2). Therefore, when the pressure changes, the equivalent capacitance C changes and the resonant frequency of the sensor f changes accordingly. The ambient pressure parameters can be sensed by monitoring the shift of the resonant frequency of the sensor.
(2)$$f=\frac{1}{2\pi \sqrt{\mathrm{LC}}}$$

### 2.2. Design and Simulation of Sensor

For the pressure sensor to have a higher Q value and better performance, we simulated and optimized the main parameters of the sensor by using HFSS software (CSRR outer diameter length a, opening width g, CSRR spacing s, CSRR width b), as shown in Figure 3. We know that there is an inseparable relationship between impedance matching and the return loss of the sensor. The smaller the S11 value, the better the impedance matching of the sensor. Therefore, we can optimize the sensor by adjusting the size of the S11 value. When the outer diameter length of the CSRR becomes longer, the resonant frequency of the sensor shifts to the lower frequency, as shown in Figure 3a. According to the formula C_p_ = 4a × Cpul (Cpul is the capacitance of the unit length of the CSRR), with a larger outer diameter length of the CSRR, the capacitance C_p_ is higher and the harmonic frequency of the sensor decreases. Figure 3b shows the resonant frequency changes with the opening spacing g of the CSRR. Cpul increases with decreasing g, resulting in an increase in total capacitance C_p_ and a decrease in the resonant frequency. When the gap s or the width of CSRR b between the outer ring and the inner ring increases, the resonance frequency gradually decreases (Figure 3b,d). The main reason is that b or s affects the capacitance and inductance of the whole structure. Increasing b or s will increase the mutual inductance capacitance and mutual inductance coefficient, and resonance frequency will also decrease. The optimized parameters of HFSS are shown in Table 1.

Figure 4 shows the electric field distribution of the sensor. We can see that the strongest electric field is mainly concentrated in the center of CSRR. When the CSRR structure is deformed, the CSRR electric field is more vulnerable to disturbance. This way, we placed the seal cavity inside the substrate and directly below the CSRR structure to sense the pressure change. When the external pressure changes, the resonant cavity will deform accordingly, resulting in a change in the resonant frequency of the sensor. The external pressure value can be calculated by extracting the resonant frequency of the sensor.

In order to better simulate the relationship between the resonant frequency of the sensor and the ambient pressure, we observe the change in the resonant frequency of the sensor by adjusting the height of the cavity. Sensor output curves corresponding to different cavity heights are shown in Figure 5a. Figure 5b shows the resonance frequency of the response curves corresponding to the sealed cavity with different heights and the relationship curve between the resonance frequency and the cavity height h2. The simulation results show that the resonant frequency of the sensor changes linearly with the cavity height, and can be used for pressure detection.

### 2.3. Sensor Fabrication

The 99% HTCC green tape (50 × 120 mm) with 130 μm thickness of single layer are used to manufacture the sensor substrate. Then, the laser drilling machine is used to drill the green tapes (the 1st and 2nd layers), the bottom substrate (the 5th and 8th layers), and the cavity structure (the 3rd and 4th layers) according to the design drawings. Double carbon films (single layer thickness is 125 μm) are filled into the square cavity structure of the third and fourth layers. Then, eight layers of HTCC green tape are precisely laminated for 10 min at 20 Mpa and 75 °C to form a physical bonding. In order to produce a strong chemical bond between the green tapes to increase the strength of the substrate, the laminated sensor is sintered in a high-temperature environment. The sintering curve is shown in [29]. When the temperature rises to 600−700 °C, the oxygen in the air reacts with the carbon film to form carbon dioxide (CO_2_), which is released from many pores of the sintered green tapes. At the same time, air from the open environment enters the cavity to maintain the pressure balance in the cavity. In order to prevent the collapse of the sensitive film, this stage is set for 300 min to make the carbon film completely burn and form a closed chamber. When the temperature rises to 1500 °C, the chemical adhesive in the HTCC green tapes is completely discharged and the ceramic particles are rearranged until they fuse with each other, finally forming an HTCC substrate with a sealed cavity. The metal electrodes on the upper and lower surfaces are printed on both sides of the substrate with silver paste using screen-printing technology. Subsequently, the printed sensor was post-fired in a muffle furnace according to the sintering profile shown in [29]. The fabricated sensor is shown in Figure 6b.

## 3. Measurement and Discussion

To characterize the performance of the fabricated sensor, we built a temperature-pressure test platform, as shown in Figure 7, which consists of high-temperature pressure furnace with pressure controller, network analyzer (N5061B, Santa Clara, CA, USA), and pressure sensor. The high-temperature pressure furnace provides high-temperature pressure environment for real-time test. The one end of the sensor with CSRR is placed in the high temperature environment of the pressure furnace, and the other end connected to the network analyzer is placed in the cooling section of the pressure furnace through the thermal insulation layer. The network analyzer is used to collect the signal reflected by the sensor. The resonance frequency of the sensor can be known from the frequency curve fed back from the network analyzer, so as to monitor the external pressure parameters in real time.

The temperature and pressure applied to the pressure sensor can be accurately controlled using high-temperature pressure test platform. The pressure sensor must be measured in the pressure range of 10−300 kPa at intervals of 50 kPa at different temperatures of 25−500 °C. At each temperature and pressure, the network analyzer will output the corresponding frequency curve, and its negative peak value is the resonance frequency point of the sensor in this environment. Figure 8a shows the sensor measured results in a pressure range of 10−300 kPa at 25 °C. As shown in Figure 8a, when the pressure increases, the resonant frequencies of pressure sensor drift toward the lower frequency. This is because the increasing pressure causes the equivalent capacitance C to increase, resulting in a decrease in the resonant frequency. The resonant frequency of the sensor decreases from 1.523 to 1.483 GHz as the pressure increases from 10 to 300 kPa at 25 °C. The resonance frequency is shifted by 40 MHz in the range of 10−300 kPa. Figure 8b shows the measured results of sensor in a pressure range of 10–300 kPa at 500 °C. Within the same pressure range, the resonant frequency of the sensor decreases from 1.502 to 1.446 GHz, and the total deviation reaches 56 MHz. This can be caused by the thermal expansion coefficient of HTCC substrate. The HTCC substrate is less hard under high-temperature environment, and the deformation is larger in the same pressure range, leading to a larger frequency offset. We have measured the sensor in high temperature and high atmospheric pressure environment for three cycles, and the measured data remain almost unchanged. Then, the three-cycles resonant frequency points under a pressure range of 10−300 kPa and at a temperature range of 25−500 °C are subsequently extracted and the corresponding linear fitted curves at different temperature are plotted in Figure 8c. The pressure sensitivity of the sensor is 139.77 kHz/kPa at 25 °C. As the temperature increases, the pressure sensitivity of the sensor increases. At 500 °C, the pressure sensitivity of the sensor is 191.97 kHz/kPa. This is because the temperature rise softens the HTCC substrate, increases the deformation distance of the cavity under the same pressure, and increases the resonance frequency shift of the sensor, resulting in an increase in the pressure sensitivity of the sensor. Furthermore, the resonant frequency points at 10 kPa and in a temperature range of 25−500 °C are subsequently extracted and the corresponding linear fitted curves at different temperature are plotted in Figure 8d. Therefore, the temperature sensitivity of the pressure sensor is 44 kHz/°C.

Figure 8c only shows the pressure sensitivity at the specified temperature (25 °C, 100 °C, 200 °C, 300 °C, 400 °C, 500 °C). In practical applications, to achieve accurate measurement of pressure parameters, it is necessary to know the pressure fitting curve at any temperature. For this, we extract the intercept and slope of the fitting curve in Figure 8c under the pressure range of 10−300 kpa and a temperature range of 25−500 °C, and plot the polynomial fitting curve of the relationship between the intercept and slope with temperature, as shown in Figure 9a. In this way, the pressure sensitivity fitting curve at any temperature can be obtained, which can be expressed as:*f_p_*(T) = intercept(T) + slope(T) × P

The error between the measured pressure and accurate pressure is plotted in Figure 9b. It can be seen from the results that the error of measured error values is relatively small under high-pressure environment (100−300 kPa). Therefore, the polynomial fitting is more suitable for accurate extraction of pressure signals in high-temperature and high-pressure environment.

## 4. Conclusions

This paper proposes a CSRR-loaded microwave SIW sensor for high-temperature pressure detection. A CSRR structure-integrated SIW resonator working in the 1.6 GHz was achieved by optimizing the sensor parameters using HFSS simulation software. Subsequently, according to the optimized parameters, the sensor was customized and fabricated on HTCC substrate using three-dimensional cofired technology and screen-printing technology. Furthermore, the pressure sensor was tested in the high-temperature pressure furnace (10−300 kPa) and can work up to 500 °C. The pressure sensitivity is 139.77 kHz/kPa at 25 °C, and with increasing temperature, the sensitivity increases to 191.97 kHz/kPa at 500 °C. The temperature compensation algorithm is proposed to achieve accurate acquisition of pressure signal in a high-temperature environment. In the future, the sensor combined with 3D printing technology and wireless transmission method will have broad application prospects in the high-temperature fields.

## Figures and Tables

**Figure 1 micromachines-14-00635-f001:**
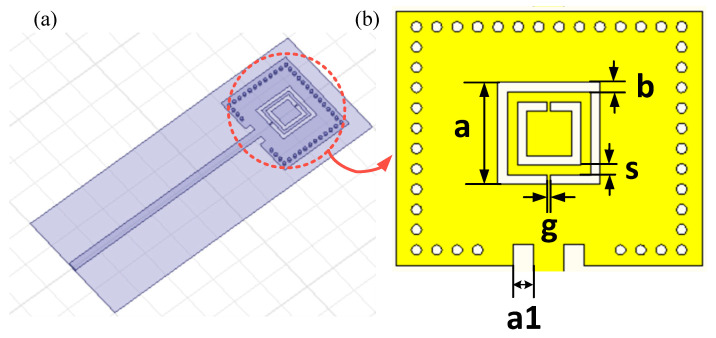
Model of the proposed CSRR integrated sensor. (**a**) Simulation model in HFSS; (**b**) physical and geometric parameters of the CSRR integrated pressure sensor.

**Figure 2 micromachines-14-00635-f002:**
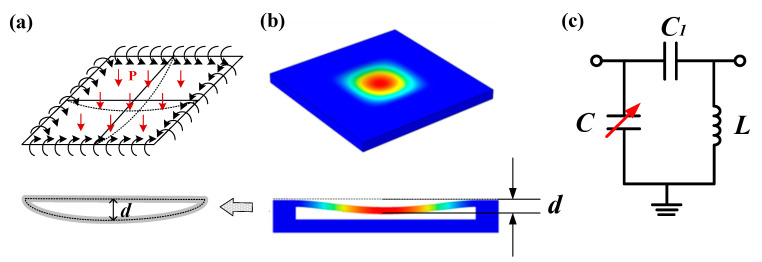
Sensor pressure detection mechanism. (**a**) The mechanical model with edges-fixed rectangular cavity with uniformly distributed load. (**b**) The stress distribution in COMSOL. (**c**) The equivalent circuit of the sensor.

**Figure 3 micromachines-14-00635-f003:**
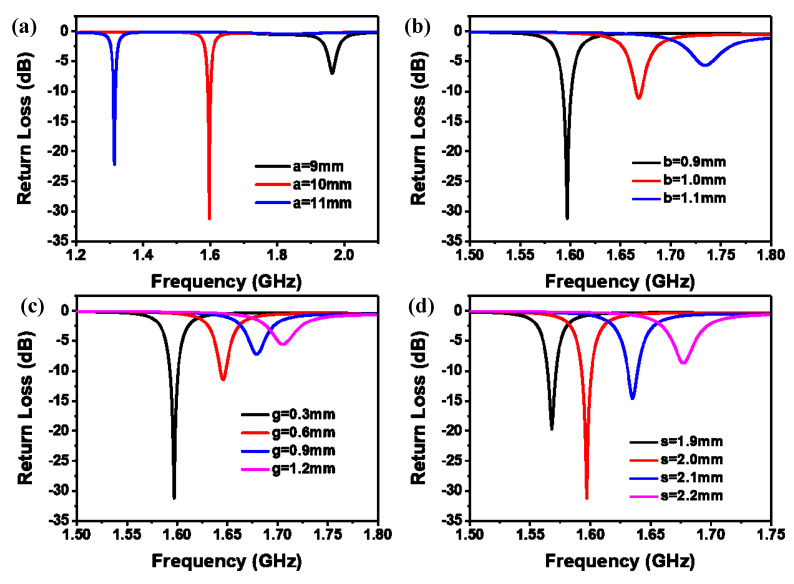
HFSS simulation optimization results. (**a**) CSRR outer diameter length a. (**b**) CSRR width b. (**c**) Opening width g. (**d**) CSRR spacing s.

**Figure 4 micromachines-14-00635-f004:**
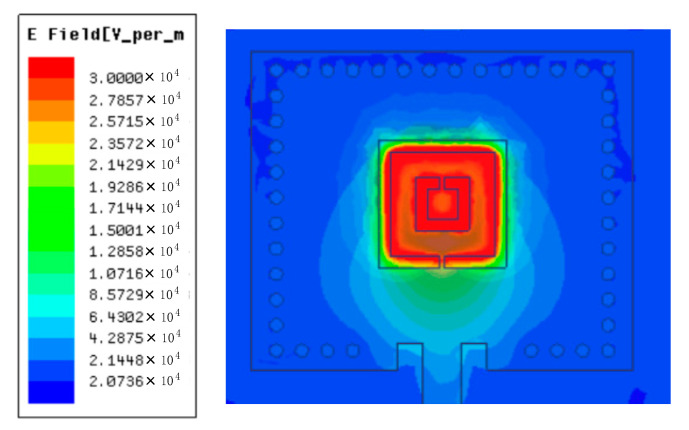
Simulated magnitude electric field distributions of CSRR-based pressure sensor.

**Figure 5 micromachines-14-00635-f005:**
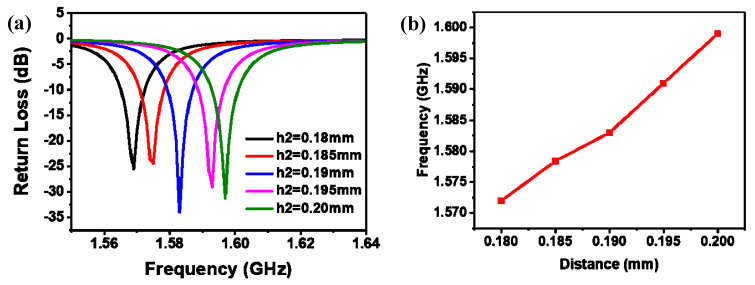
HFSS simulation optimization results. (**a**) The output curves corresponding to different cavity heights; (**b**) the variation curve of resonance frequency with cavity height.

**Figure 6 micromachines-14-00635-f006:**
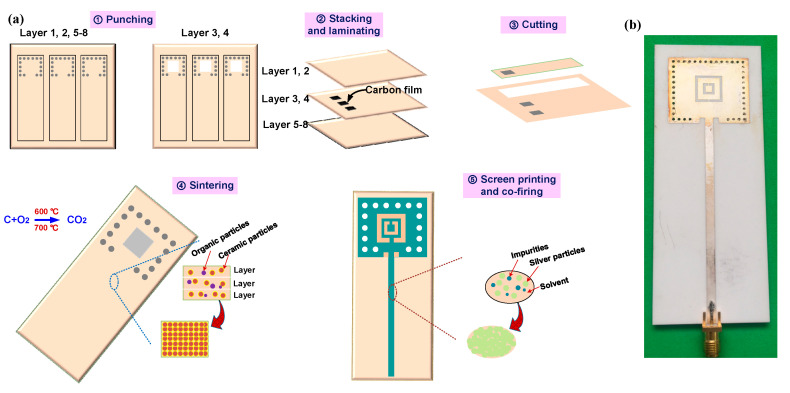
(**a**) The process of sensor fabrication; (**b**) the fabricated sensor.

**Figure 7 micromachines-14-00635-f007:**
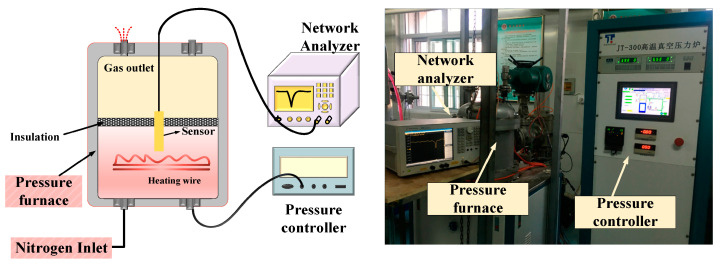
High-temperature pressure measurement platform.

**Figure 8 micromachines-14-00635-f008:**
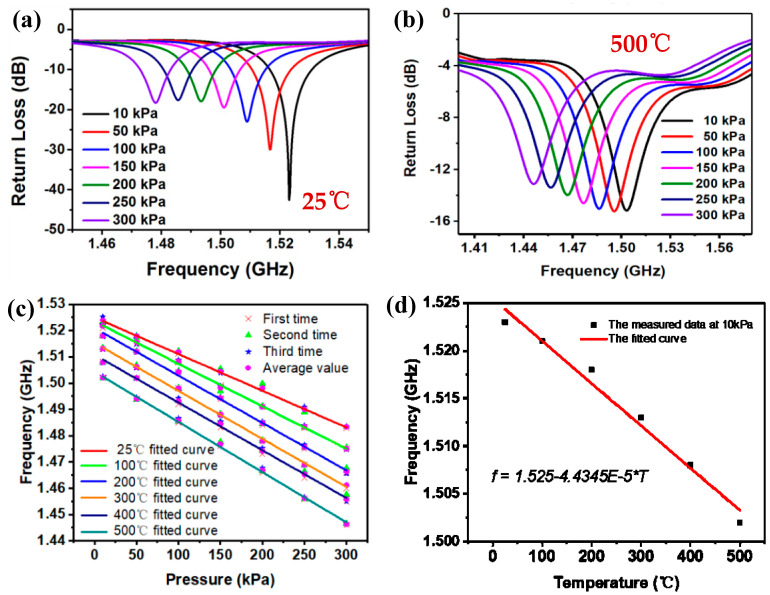
(**a**) Resonant frequency curves within the pressure of 10–300 kPa at 25 °C; (**b**) resonant frequency curves within the pressure of 10−300 kPa at 500 °C; (**c**) linear fitting curves of pressure at 25−500 °C; (**d**) linear fitting curves of temperature at 10 kPa.

**Figure 9 micromachines-14-00635-f009:**
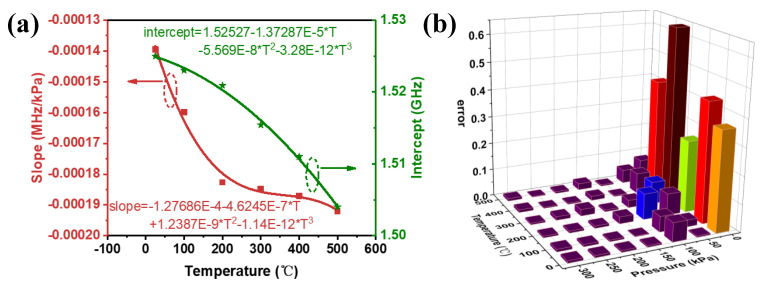
(**a**) Polynomial fitting slopes and intercepts for curves of pressure versus resonant frequency under different temperatures.; (**b**) nonlinearity errors in the pressure range of 10−300 kPa at 25−500 °C.

**Table 1 micromachines-14-00635-t001:** The detailed parameters of the designed sensor.

Parameter	Value (mm)
a	10
b	1
s	2
g	0.3
a1	2

## Data Availability

Not applicable.

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
