# Peer review of "A Microwave Pressure Sensor Loaded with Complementary Split Ring Resonator for High-Temperature Applications"

_micromachines, 2023, doi:10.3390/mi14030635_

Round 1

Reviewer 1 Report

The authors have presented the outcome of their research on a high-temperature pressure sensor based on a SIW based CSRR. The SIW based pressure sensors operating are not really new. At least, one of the authors of this work has published this work before: Hairong Kou et al 2020 J. Phys. D: Appl. Phys. 53 085101.  The CSRR is largely similar to the one published two years ago.  Even the substrate materials are the same.  

Hence,  I would like to raise the following issues:

1) In figure 8d, there is NO suggestion that, over such a wide temperature range, the measured correlation between the resonant frequency and pressure is completely linear.

2) The major difference this work and the previous work apparently lies in the temperature range and the resonant frequencies. I am not aware of any other significant difference. Perhaps, the authors may highlight all the differences between this work and the one previously published in IOP.

Reviewer 2 Report

In the paper, the author has proposed a passive substrate integrated waveguide (SIW) pressure sensor based on CSRR, which realizes sensing applications in high-temperature environments at a microwave frequency band by using the electromagnetic coupling principle. A small sealing cavity, prepared by the HTCC three-dimensional integrated manufacturing process, is placed under the CSRR. When the sealing cavity is deformed under pressure, the electromagnetic field close to the CSRR resonator is disturbed, resulting in the shift of the resonant frequency. By analyzing the return loss of the resonant frequency of the sensor, the relationship between the environment and the sensor can be obtained. The paper is well executed and deserves publication after revisions. I have the following suggestions

1) In the title of the paper, the full form of CSRR should be given instead of the abbreviation. 

2) Kindly mention the version of the COMSOL software which has been used in line 104. Also, provide the boundary condition, excitation port, mesh size, and other important settings. 

3)At first the author used COMSOL, then switched to HFSS software. Kindly explain the need for this transition, and why it can't be simulated in one software. 

4) What is the reproducibility and lifetime of this sensing device?

5) Introduction section should be modified by adding information on other types of pressure sensors such as polymer waveguide-based mechanical sensors. And provides the disadvantages of those sensors to the one proposed in this work as polymers can not withstand high temperatures. 

6) There are several English grammar-related errors that need to be corrected throughout the paper. 

7) The author should state the factors or parameters that can be modified to enhance the sensitivity of the device.

8) Is 500 degrees the limit of the sensor material before melting? How temperature resistance of the sensor can be further enhanced?

Author Response

Please see the attchment.

Round 2

Reviewer 2 Report

The author has satisfied the reviewer's concerns. I am willing to accept the paper in its current form. 

Author Response

  We feel grateful and pleased that you offer valuable comments for our research work (micromachines-2242601 A microwave pressure sensor loaded with CSRR for high-temperature applications). We have substantially revised our manuscript after your comments. All revisions in manuscript were marked in RED font.

If you have any question about this paper, please don’t hesitate to let us know. We look forward to hearing from you soon for a favorable decision. Our research team feels thankful about your comments for our research work and wish you have a nice day!

Best regards,

Hairong Kou

[email protected]
